# No Recommendation Is (at Least Presently) the Best Recommendation: An Updating Quality Appraisal of Recommendations on Screening for Scoliosis

**DOI:** 10.3390/ijerph19116659

**Published:** 2022-05-30

**Authors:** Maciej Płaszewski

**Affiliations:** Faculty of Physical Education and Health in Biała Podlaska, Józef Piłsudski University of Physical Education, 00-809 Warsaw, Poland; maciej.plaszewski@awf.edu.pl

**Keywords:** screening for scoliosis, guidelines, standards of development, strength of recommendations, levels of evidence, expert evidence

## Abstract

Recommendations addressing screening for scoliosis differ substantially. Systematically developed guidelines are confronted by consensus and opinion-based statements. This paper elaborates on the issue of the standards of development and reporting of current guidelines and recommendation statements, as well as on the methodological quality of the corresponding evidence syntheses. The SORT classification and the AMSTAR 2 tool were used for those purposes, respectively. Publications included in the analysis differed substantially in terms of their methodological quality. Based on the SORT and AMSTAR 2 scores, the 2018 US PSTF recommendation statement and systematic review on screening for scoliosis are trustworthy and high-quality sources of evidence and aid for decision making. The recommendation statement on insufficient evidence to formulate any recommendations is, paradoxically, very informative. Significantly, updated opinion-based position statements supporting screening for scoliosis acknowledged the importance of research evidence as a basis for recommendation formulation and are more cautious and balanced than formerly. Expert opinions, not built on properly presented analyses of evidence, are at odds with evidence-based practice. Nonetheless, contemporary principles of screening programs, especially those addressing people’s values and preferences, and the possible harms of screening, remain underrepresented in both research and recommendations addressing screening for scoliosis.

## 1. Introduction

### 1.1. Rationale

Adolescent idiopathic scoliosis, AIS, is a health condition that affects 1–3% of the general population [1,2]. Potential serious physical and/or psychosocial consequences, such as pain, pulmonary deficiency, as well as disturbed body image and mental health, are not necessarily correlated with the magnitude of deformity [2,3,4,5]. Treatment effectiveness, including conservative and orthopaedic interventions, remains a matter of long-lasting debate [2,3,4,6]. Another disputed issue is whether early detection of mild scoliosis, which may lead to early, thus possibly effective, treatment, should be done through clinical examination or a population screening, typically school screening [3,6,7,8]. Screening programs may be beneficial, but they may also be harmful [9,10,11], and this may well apply to screening for scoliosis [4,12,13].

School screening for scoliosis applies to millions of schoolchildren globally [3,8,13,14], in their unique and fragile time of puberty [15]. National and regional policies range from obligatory to discontinued screening programs, and recommendations vary from encouraging, through no recommendations due to insufficient evidence to formulate recommendations, to recommendations against screening [10,11,12,13,14,15,16,17,18,19,20].

One important, and long debated issue is the X-ray exposure and related long-held view of carcinogenic risks from X-rays. Current (2021) UK National Screening Committee, UK NSC, recommendations against screening for scoliosis include potential overdetection, waste of resources, and unnecessary X-ray exposure, as reasons for not meeting the criterion of “a simple, safe, precise and validated screening test”, needed to justify a screening program [20]. In the research synthesis study prepared for the US Preventive Services Task Force, US PSTF guideline formulation (2018) [3] no studies met inclusion criteria to answer the key question “What are the harms of screening for AIS?”. On the other hand, routine paediatric radiographic scoliosis screening is regarded by some experts as safe and beneficial in early and accurate diagnosis, in opposition to “non-radiographic screening methods” [21]. Introducing modern devices, producing low, or ultra-low dose radiation (e.g., [22]) supports such opinions. Importantly, however, harms are no longer understood purely in terms of direct adverse events or side effects of diagnostic testing, such as X-ray exposure. Health lost due to the overdiagnosed or false-positive health state, such as anxiety and complications of labelling, diagnostics, unneeded or unnecessary treatment, and stigmatization, are considered as harms [9,11,23].

Guidelines and recommendations not only remain discrepant or contradictory but were developed according to various standards and procedures [16,20,24,25,26]. The evolution of clinical practice guidelines into systematically developed, evidence-based documents, instead of consensus-based statements, serve for improving health outcomes [27,28,29,30]. Trustworthy guidelines and recommendations are developed through rigorous methodological standards, with the certainty of evidence and strengths of recommendations as key features [24,25,27,30,31]. Several rating systems have been introduced for those purposes, initially addressing treatment, then other interventions, such as preventive services and screening [27,28,29,30,31,32,33]. Opinion-based recommendations are on one hand regarded as invalid [27,28,34,35]. On the other hand, however, evidence-informed consensus statements and expert evidence are acknowledged and regarded as needed and unique, especially in providing new ideas and insights [10,36,37,38,39,40]. Therefore, it is important to distinguish trustworthy from potentially misleading recommendations. And it is not merely a matter of technical appraisal of the methodological quality of their development. Issues such as distinguishing between expert opinions and expert evidence, the importance of narrative reviews in formulating new ideas and informed opinions, as well as contemporary understanding of screening programs, highlighting problems such as harms of overdiagnosis and unnecessary treatment, imply that the distinction between evidence-based and opinion-based recommendations is more complex than just the distinction between methodologies of their development.

### 1.2. Objectives

The study addresses the trustworthiness of current recommendations and position statements addressing screening for scoliosis in terms of their methodological quality and standards of development. Strengths of recommendations and methodological quality of the underlying evidence reviews were assessed.

Consequently, the paper updates earlier research synthesis studies, conducted in 2011 [16] and 2013 [26]. Since that time, major recommendation statements have been updated and/or revised [17,18,19,20] so that an updating appraisal was warranted.

This report also provides a methodological enhancement of another analysis [13], focused on whether and how the contemporary standards of screening programs, specifically issues of person-centeredness and harms of screening interventions, are represented in current recommendations and policy statements on screening for scoliosis.

## 2. Materials and Methods

### 2.1. Study Conduct and Reporting

This study was informed by the PRISMA 2020 guideline for reporting systematic reviews, specifically as regards guidance for conducting and reporting updated reviews [41]. Nevertheless, the study, as this was not a formal systematic review, was also informed in terms of the principles and methods of narrative reviews [42].

### 2.2. Steps of the Study

The analysis comprised two stages. Firstly, the current recommendations on screening for scoliosis were examined in terms of their standards of development. For that purpose, their content was analyzed according to the Institute of Medicine criteria for guideline development and reporting (Table A1) [28]. The strengths of recommendations and levels of evidence were assessed using the Strength of Recommendation Taxonomy, SORT, classification (Table A2) [30]. Secondly—which applied for guidelines based on systematic reviews of evidence—a quality appraisal of evidence reviews that had been used in the process of recommendation formulation, was conducted using the revised A MeaSurement Tool to Assess Systematic Reviews, AMSTAR 2 tool (Table A3) [43]. The steps of the study are presented in Figure 1. Details of the assessment methods are described in Section 2.5.

### 2.3. Eligibility Criteria

#### 2.3.1. Guidelines and Recommendation Statements

All current (i.e., new or updated, and not specified as outdated, withdrawn, or retired) recommendations and position statements, regarding screening for scoliosis, were included in the initial analysis. Both opinion-based and evidence-based documents were eligible. New or updated publications were eligible if they were not included in the preceding appraisals [16,26]. Thus, papers published before 2014 were not considered eligible.

#### 2.3.2. Review Reports

The starting point to define the types and forms of review studies was the classical characteristics of systematic review (such as that in the glossary of terms from the Epistemonikos database: “*A review of a clearly formulated question that uses systematic and explicit methods to identify, select, and critically appraise relevant research, and to collect and analyse data from the studies that are included in the review. Statistical methods (meta-analysis) may or may not be used to analyse and summarise the results of the included studies*” [44]). Nonetheless, to address the issue more adequately and comprehensively, a broader understanding and usage of meta-research [45,46,47], as well as the shift from “*systematic review*” to “*evidence synthesis*” [3] were incorporated.

Review reports prepared for the development of guidelines (either commissioned or conducted by the guideline development group) were included. External systematic reviews, referred to in the analysed guidelines, but not prepared in the process of their development, were not included in the analysis of methodological quality with AMSTAR 2, but were only considered in the assessments of strengths of recommendations with SORT.

### 2.4. Identification of Relevant Publications

PubMed and pre-appraised sources—NICE Evidence, TRIP, PEDro, and REHAB+ databases, as well as G-I-N and INAHTA guideline registries, were searched between 5 and 8 February 2022 for new or updated pertinent publications, published since January 2014 (i.e., to the latest date of publication applied in the previous study [26]). As this was an updating search, previously employed search terms [16,26] were used. In PubMed the MeSH terms “scoliosis” and “mass screening” as well as the free terms “scoliosis” and “screening” were used together with the filters “systematic review” and “meta-analysis”. PubMed “*Similar articles*” and “*Cited by*” navigation tools were used for further browsing. For other bibliographic databases, corresponding search terms and limits were used. Additionally, the reference lists of papers included for full-text analysis as well as websites of institutions that have released the included documents, were checked for information on updates, discontinuations, or withdrawals. The PRISMA 2020 [41] flow diagram for updated systematic reviews, presenting the updating search and selection process, is presented in Figure 2.

### 2.5. Appraisals and Quality Assessments

#### 2.5.1. Guidelines and Recommendation Statements

Clinical practice guidelines are “*systematically developed statements aimed at helping people make clinical, policy-related and system-related decisions*” [27]. In the first step, to classify and distinguish the initially included documents as evidence-based, systematically developed guidelines or opinion-based statements, the relevant characteristics of evidence-based guidelines from the Institute of Medicine were applied [28] (Table A1).

The SORT classification was then used to assess the strengths of recommendations and levels of evidence of the included documents [30]. SORT taxonomy is based on three criteria: the level (quality) of evidence, whether the evidence is patient-oriented, and the consistency and coherency across studies (Table A2).

#### 2.5.2. Evidence Syntheses

The methodological quality of the included systematic reviews was assessed with the AMSTAR 2 tool [43]. The AMSTAR 2 comprises 16 criteria for critical appraisal of systematic reviews of studies of healthcare interventions (Table A3). In contrast to the original AMSTAR tool, the AMSTAR 2 was developed to enable appraisals of systematic reviews of randomised trials, but also of non-randomised studies, or both. The instrument has an overall rating, based on weaknesses in critical domains, rather than on an overall score. Overall confidence in the results of a systematic review is rated by considering the potential impact of an inadequate rating for each item and assessing a review in terms of critical weaknesses and critical flaws. Noticeably, appraisers are to decide (judge) which items are most important for the assessed reviews [43].

Appraisals were done in duplicate, with a collaborator, and uncertainties were resolved through discussion.

## 3. Results

### 3.1. Updating Search Results

Four updated recommendation documents have been published since 2014 [17,18,19,20], and they were included for analysis. Two of the recommendation statements were supported by evidence syntheses (systematic reviews) [3,48]. They were therefore also considered eligible. Thus, four recommendation documents [17,18,19,20] and two systematic reviews [3,48] were deemed eligible (Figure 2, Table 1).

In 2021, UK NSC published a commissioned evidence map that aimed at establishing whether an updating evidence review should be conducted based on available evidence for purpose of updating their scoliosis screening recommendations [8]. Out of 622 publications assessed for eligibility, only one primary study, with “*purely speculative conclusions*” [8], was considered potentially relevant. On that basis, the UK NSC concluded that the volume and type of evidence were at that time insufficient to justify an evidence review [20]. Therefore, as the UK NSC have not released updated recommendations, their 2016 recommendation document, as current for 2021 [20], as well as the background systematic review published in 2015 [48], were included in the present analysis. Accordingly, the 2021 evidence map [8] was excluded.

Two more systematic reviews addressing screening for scoliosis were published after 2014 [14,49]. Nonetheless, quality appraisals of individual evidence syntheses, which were not supporting any of the included guidelines or recommendation statements, were beyond the scope of this study. Therefore, those reports were excluded from the analysis (Figure 2).

### 3.2. Quality of Guideline Development

The analysis of current guidelines and recommendation statements according to the Institutes of Medicine standards of guideline development and recommendation formulation is presented in Table 2.

### 3.3. Strengths of Recommendations and Levels of Evidence

The SORT qualifications of the identified recommendation documents are presented in Table 3. The systematically developed recommendations (US PSTF [19] and UK NSC [20] were ranked higher in SORT than consensus-based (SOSORT [18]) and opinion-based (SRS/AAOS/POSNA/AAP [17]) statements.

### 3.4. Methodological Quality of Evidence Reviews

Table 4 summarises individual AMSTAR 2 assessments. The overall rated confidence in the results of the systematic review conducted in the preparation of the US PSTF [3] and the UK NSC [48] recommendation were high and moderate, respectively (Table 4).

## 4. Discussion

The results of this study confirm that systematically developed, evidence-based recommendations, when derived in the process of a thorough, procedural development (Table 2 and Table 3), are of greater trustworthiness than consensus-based recommendations (Table 2), even if the latter ones are more conclusive (Table 1).

Nonetheless, the differences are not as clear as they appear, and the problem matter is not that straightforward. The value of **expert evidence**, the importance of **scholarly narrative reviews**, as well as **contemporary understanding of screening programs**, set a different light on consensus-based and opinion-based recommendations. And the distinction between ‘trustworthy’ and ‘flawed’ statements is not just parallel to the methodological approach of guideline development. Therefore, both the results of this study and the above-mentioned problems are discussed below.

### 4.1. Quality of Evidence Analyses

#### 4.1.1. Strength of Recommendations and Levels of Evidence

The Scottish Intercollegiate Guideline Network, SIGN Grading System 1999–2012, applied in the previous analysis of guidelines and recommendations [16], was subsequently withdrawn from use for guideline appraising [31]. Therefore, in this updating analysis, the SORT [30] classification was used. It is a recognised [50] and currently used [51,52,53,54] measure, addressing person-centred (patient-oriented) outcomes, as well as the quality, consistency, and coherence of the underlying evidence, in terms of the strength of recommendations (Table A2).

#### 4.1.2. Appraisals of Review Studies

In contrast to the original AMSTAR, the AMSTAR 2 was designed to be usable for the assessment of systematic reviews of randomised trials, as well as systematic reviews of non-randomised studies [43]. This is advantageous, considering the inconsistent and low-quality evidence base addressing screening for scoliosis [3,8,54]. AMSTAR 2, in contrast to the original tool, is not intended to generate an overall score for an appraised paper.

Nonetheless, 10 out of 11 original domains are retained [43]. Therefore, assessing additional reviews with AMSTAR 2 will not exclude building on the previous evidence syntheses (using the original AMSTAR) [26,54].

### 4.2. Expert Evidence versus Expert Opinion

There is a tendency to view guidelines that use the term *“expert opinion”* as less rigorous [27,28,29,31,37]. Guidelines are considered evidence-based when the evidence underpinning recommendations is identified, selected, appraised, synthesised, and presented systematically and transparently [24,25,27,34,39]. Nonetheless, at least at the stages of the assessment of the overall body of evidence and critical appraisal, **some degree of personal judgment**—purposeful or unconscious—**is inevitable [33,36,37]**. **The opinion is not evidence but experts are needed**. Multidisciplinary guideline development teams are required for modern guidelines, with experts needed to help understand the evidence as well as to represent patient voices and stakeholders’ needs [27,28,29,37,39,43].

Sometimes using evidence systematically and transparently can be challenging; in some circumstances (such as unpublished or indirect evidence, missing contextual information), experts may be the only or main source of evidence [37,39,40]. To use expertise wisely in the context of evidence, a **distinction between expert evidence and expert opinion is proposed**, with particular importance of this distinction in terms of guidelines robustness and trustworthiness [37,40].

People’s views, preferences, and circumstances, as well as person-centeredness, are contemporarily acknowledged and promoted in health policies, including concepts and principles of screening [9,10,11,13]. Despite that, contextual factors and person-centred outcomes are still very much underrepresented in scoliosis literature [3,5,13,55]. Opinions of experts can be misleading and dangerous [56], and subsequent screening and treatment policies harmful [4,9]. However, if properly formulated and used [37,39,40], may also be valuable and indicatory [12,18,57,58,59].

Expert opinions were appreciated in the Scoliosis Research Society, SRS, International Task Force’s recommendation formulation process, prepared through a methodological study, as of particular importance for considering perceived harms in the absence of available research evidence on this critical issue [57]. Correspondingly, contemporary standards and policies regarding specific needs and circumstances of adolescents, including careful spine examinations during individual adolescent Periodic Health Visits, not population-based school screenings, are formulated in the Bright Futures expert panel guidelines of the American Pediatric Society [7].

### 4.3. The Value of Scholarly Narrative Reviews

Systematic reviews are at the top of the classical pyramid hierarchy of evidence, whilst narrative reviews are at the bottom [25,30]. They tend to be disregarded as lacking an explicit intent to maximize scope or analyse data collected, with conclusions that they may reach therefore being prone to bias, with literature selection in a way that supports the authors’ view rather than on the basis of evidence [40,45,47]. Some recent classification approaches do not even consider narrative reviews, as lacking explicit rigorous methods of conduct [46]. On the other hand, the rigid distinction between systematic and narrative (opinion) reviews is criticised as spurious [40] by authors who once disseminated the principles of Evidence-Based Medicine, including hierarchies of evidence [12,42], and who promote the developments of this movement [59,60].

Thus, systematic and narrative reviews serve different purposes and should be viewed as complementary. While conventional systematic reviews address narrowly focused (typically population-intervention-comparator-outcome, PICO) questions, and their key contribution is summarising data, narrative reviews provide interpretation and critique, and their key contribution is to deepen understanding. In that sense, a narrative review is **a scholarly summary along with interpretation and critique**, which can be conducted using a number of **distinctive methodologies, differing from the classic methodology of systematic review, but not unsystematic** (i.e., careless or conducted ad hoc) [25]. Narrative reviews emphasize thoughtful, in-depth, critically reflective processes—thinking and interpretation, and thus are **evidence-informed rather than evidence-based** [25,46,60,61].

### 4.4. New Understanding and Terminology of Screening

Not only the issues of expert evidence [39,40], contemporary approaches to narrative reviews [42], and evidence-informed practice [61] put a different light on the dilemma of the opposing systematically developed and opinion-based recommendations on screening for scoliosis.

Screening principles, and also terminology, have evolved, and crucial proposals, such as shifting from **screening for disease to the prediction of poor health, from early disease detection to preventive service,** and from asymptomatic subjects to **otherwise healthy individuals**, have been formulated [9,10,11]. Analyses of the issue of discrepant and confusing scoliosis screening recommendations, led by the author of this study, also shifted from data analyses and critical appraisals of the underlying evidence [16,26,54] to more person-centred as well as narrative expert evidence processes [5,13].

### 4.5. When No Recommendation Is the Best Recommendation

The US PSTF have changed their recommendation from “*against”* screening for scoliosis (1996, 2004) [16,26] to “*no recommendation”* due *to “insufficient evidence to assess the balance of benefits and harms of screening*” [19]. Surprisingly, the former (2004) statement was based on a low-quality brief evidence synthesis [16], while the current update is a high-quality document, following modern and demanding standards of guideline development [24,25], and based on an exhaustive evidence synthesis study [3] (Table 3).

The “I” recommendation is addressing the body of evidence (“*insufficient evidence*”) in terms of recommendation formulation of whether to screen or not to screen for scoliosis. The decisions are, however, left to decision-makers—patients and clinicians [19]. Consequently, the recommendations were criticised as inconclusive and indecisive [58,62].

On the contrary, two arguments support the sense of, and need for, conditional recommendations or statements of no recommendation, in the presence of insufficient or ambiguous evidence: (1) when evidence is sparse or doubtful (insufficient), recommendations need to be issued; otherwise, guideline panels miss an opportunity to support clinicians and patients [40] (2) conditional recommendations based on explicit evidence analyses encourage further relevant research. “*Clinical guideline panels should have the courage to make statements of no recommendation if the evidence base is weak”. Such a recommendation could actually be the best guidance, and give impetus to important research”* [63]*,* as the underlying low-quality evidence can be counterproductive or even harmful.

The US PSTF “I” recommendations on important health topics are annually reported to the US Congress. Adolescent health topics, including mental and behavioural health, were reported in 2020 [64]. The need for further research was acknowledged in response to the updated US PSTF recommendations [12,58].

### 4.6. Compelling Need for Further, People-Centred Research

The Oxford Centre for Evidence-Based Medicine, OCEBM, levels of evidence classification, with questions referring to early detection (screening) tests (added in the revised version of the classification as *“sufficiently important to merit a separate entry”* [33]), gives a good perspective of the most desired and eligible study designs for the questions on evidence on screening tests: preferably systematic reviews of randomised controlled trials, less preferably observational studies, then mechanistic reasoning, and not expert opinions formulated without referring to pre-appraised evidence (Table A4).

Additionally, considering contemporary standards of person-centred care [13,20,40,53], as well as scarce research evidence on person-centred outcomes as regards to screening programs [3,8,48,55], **not only further, and methodologically rigorous research, but also more people-centred research is warranted**.

Based on the presented updating analyses, it is clear that the lack of an adequate evidence base remains a crucial problem in recommendation formulation of whether to screen or not to screen adolescents for scoliosis.

## 5. Conclusions

The US PSTF has issued high-quality guidance [19], and the UK NSC recommendations are also based on evidence syntheses [20]. The authors of opinion-based recommendations from professional societies (SRS/AAOS/POSNA/AAP [17] and SOSORT [18]) also acknowledge the importance of research evidence as a basis for recommendation formulation.

Nonetheless, the key issues are that recommendations remain contradictory, that an evidence base to formulate trustworthy statements remains lacking, and that the contemporary principles of screening programs, especially those addressing people’s values and preferences, and the possible harms of screening, are still underrepresented in research and recommendations.

“*All screening programmes do harm; some do good as well.*” [9]

## Data Availability

Not applicable.

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
