# Peer review of "No Recommendation Is (at Least Presently) the Best Recommendation: An Updating Quality Appraisal of Recommendations on Screening for Scoliosis"

_ijerph, 2022, doi:10.3390/ijerph19116659_

Round 1

Reviewer 1 Report

Congratulations on your thorough and high-quality paper.

Suggestion:

Section 1.1. Please add a paragraph between paragraph 1 and 2. This new paragraph can briefly outline the debated pro and cons of scoliosis screening programs. In this paragraph it would be good to mention also that X-ray exposure, a long-held view of carcinogenic risks from X-rays are even hotly debated (e.g. Oakley et al. Dose Response).

Minor typos:

Section 2.3.2, para 2. Change "As regards study selection criteria..." to "As regards to study selection criteria..."

Section 4.6, para 2. Change "...as regards screening..." to "...as regards to screening..."

Reviewer 2 Report

The present review addresses the credibility of recommendations for screening scoliosis updating the author's previous review. The manuscript presented addresses a relevant topic. I have some comments.

Introduction

Please develop the reasons to perform this review. What does this one add comparatively to the last performed, and why do you consider it worthwhile.

Methods

It should present more clearly the design of the study.

Please refine the search strategy, selection process, data collection process, and data items. We encourage the author to add a flow chart for updated reviews.

Results

Line 175   Please add the reference for the statement: study, with “purely speculative conclusions”,   

Discussion

Discuss limitations and implications of the results obtained as well as the need for future work on the issue.

Reviewer 3 Report

This is an article related to a specific issue: recommendations about screening for scoliosis. The author seems to be an expert in this field as in 2012 and 2014 he realized previous appraisals related to the literature present in this field. The author seems in general to manage well the methodology and the tools to evaluate guidelines and recommendations in health.

This work is related to an important question (to do or not to do screening for scoliosis) that can have an impact on our health system.

The title is attracting, the abstract is fair; the Rationale and Objectives are well-written, as summarizes the recent research related to the topic and explains the need of the review. The Conclusions are clear.

In my opinion, the major flaws are related to the transmission of the contents. I am an expert in the field of scoliosis but not in recommendations and guidelines. For me the reading has been difficult, I had to read many times the article because I had some problems in understanding. I think that the specificity of the treated issues does not help in the comprehension of the article and it is easier being distracted in the reading or jumping pieces of the article, looking for the main results. So, as it treats an interesting issue by a specialist in this field, in my opinion this article needs to become clearer and more attractive for the reader; also it needs a linguistic revision by a native speaker.

1. About language, I found many times hard to understand the use of two names together: expressions like recommendation formulation, recommendation preparation, etc. have to be changed.

2. Moreover: Page 3, line 84: I presume the use of against ( their content was analysed against the Institute of medicine criteria) is not appropriate. Possible a better way would be: according to. Same observation for page 5, line 186

Material and Methods

3. Could you include some graphycal picture (as a flowchart) that explains/represents the two stages or two steps of the study (1- application of Institute of Medicine criteria + SORT; 2- AMSTAR 2 tool)?

4. Identification of relevant publications.

pag 3, line 127: previously employed search terms

It could be easier for the reader to find in the article which are these terms used previously. Otherwise for lack of time or of resources, the most of the readers would not search the articles where you communicated the research.

5. pag 4, (lines 138-142). When you explain SORT, can you use some synonimous or other words? The repetition of the words of the acronimous makes hard to understand what it is. What about for example: the Strenght of Recommendations Taxonomy (SORT) is a classification applied based on 3 levels of quality: A, B and C.?

Results

6. Page 4, line 161: I think that “two of the documents” is not right, as in this way it seems they are two of the four documents previous cited. On the contrary, as I understand, these are two more documents, you should write: “two more documents”.

7. Pag 5 , from line 171 to 180. You could clarify this paragraph expliciting the year (2016) when you mention the article (33)

8. Pag 6, line 194: table 5 does not exist. Did you mean Table 3?

9. Pag 6, line 198: table 3. Did you mean table A3?

Discussion

10. At the begininning you mention you will treat the limitations of the study but I did not find them.Where do you treat this aspect? Could you be more explicit?

11. In Discussion there is given too much space to the theorical background compared to space dedicated to discuss the results: in 4.3, 4.6 scoliosis screening does not appear. Why not introduce this theorical information in Introduction?

12. pag 9, lines 326-329: it is a very important point (persone centred care and research) but the phrase is unclear or incomplete (“given contemporary standards” would be “to give contemporary standards”?)
